# Structural mechanism of protein recognition by the FW domain of autophagy receptor Nbr1

Jianxiu Zhang [1,2,7], Ying-Ying Wang[3,4,5,7], Zhao-Qian Pan[4,7], Yulu Li[4], Jianhua Sui [4,6], Li-Lin Du [4,6 ✉] & Keqiong Ye [1,2 ✉]

Neighbor of BRCA1 (Nbr1) is a conserved autophagy receptor that provides cargo selectivity to autophagy. The four-tryptophan (FW) domain is a signature domain of Nbr1, but its exact function remains unclear. Here, we show that Nbr1 from the filamentous fungus *Chaetomium thermophilum* uses its FW domain to bind the α-mannosidase Ams1, a cargo of selective autophagy in both budding yeast and fission yeast, and delivers Ams1 to the vacuole by conventional autophagy in heterologous fission yeast. The structure of the Ams1-FW complex was determined at 2.2 Å resolution by cryo-electron microscopy. The FW domain adopts an immunoglobulin-like β-sandwich structure and recognizes the quaternary structure of the Ams1 tetramer. Notably, the N-terminal di-glycine of Ams1 is specifically recognized by a conserved pocket of the FW domain. The FW domain becomes degenerated in fission yeast Nbr1, which binds Ams1 with a ZZ domain instead. Our findings illustrate the protein binding mode of the FW domain and reveal the versatility of Nbr1-mediated cargo recognition.

[1] Key Laboratory of RNA Biology, CAS Center for Excellence in Biomacromolecules, Institute of Biophysics, Chinese Academy of Sciences, Beijing 100101, China. [2] University of Chinese Academy of Sciences, Beijing 100049, China. [3] College of Life Sciences, Beijing Normal University, 100875 Beijing, China. [4] National Institute of Biological Sciences, 102206 Beijing, China. [5] School of Basic Medical Sciences, Henan University of Science and Technology, Luoyang 471023 Henan, China. [6] Tsinghua Institute of Multidisciplinary Biomedical Research, Tsinghua University, 102206 Beijing, China. [7] These authors contributed equally: Jianxiu Zhang, Ying-Ying Wang, Zhao-Qian Pan. ✉email: dulilin@nibs.ac.cn; yekeqiong@ibp.ac.cn

Autophagy transports cytoplasmic materials (cargos) to lysosomes/vacuoles and is important for maintaining cellular homeostasis[1,2]. Cargos can be transported in a non-selective or selective manner. In selective autophagy pathways, cargos are specifically recognized by autophagy receptors[3–6].

Nbr1 is an autophagy receptor conserved across eukaryotes, but notably absent in *Saccharomyces cerevisiae*. In mammalian cells, NBR1 targets protein aggregates, midbodies, peroxisomes, and focal adhesions for autophagic degradation[7–10]. In *Arabidopsis thaliana*, NBR1 mediates selective autophagy of protein aggregates and viral proteins, contributing to plant stress response[11–14]. In the fission yeast *Schizosaccharomyces pombe*, Nbr1 transports three aminopeptidases Lap2, Ape2, Ape4, and α-mannosidase Ams1 from the cytosol into the vacuole, where these proteins function as degrading enzymes[4,15]. This Nbr1-mediated vacuolar targeting (NVT) pathway in *S. pombe* depends on the endosomal sorting complexes required for transport (ESCRTs), but not the conventional autophagy machinery. In the filamentous fungus *Sordaria macrospora*, NBR1 is involved in selective pexophagy[16].

Nbr1 proteins in different eukaryotic species contain a variable number of domains. These domains are generally involved in protein-protein interaction and include the PB1 domain, the ZZ-type zinc-finger domain, the FW domain, the UBA domain, and the LC3-interacting region (LIR) motif (Fig. 1a). The PB1 domain is located at the N-terminus and mediates oligomerization[17]. The ZZ domain, present in one to four copies, recognizes N-terminal peptides and global structures of cargo proteins[15,18–21]. The UBA domain binds ubiquitin, a signal for protein degradation. The LIR motif, also called Atg8-interacting motif (AIM), interacts with LC3/GABARAP/Atg8, a key component of the conventional autophagy machinery[16,22].

The FW domain (also known as NBR1 domain), named after four conserved tryptophan residues, is a signature domain found in all Nbr1 proteins[22–24]. This domain has been annotated by protein domain databases (Pfam database ID PF16158, NCBI CDD database ID cd14947, and InterPro database ID IPR032350). Besides Nbr1 proteins, the FW domain also exists in ILRUN proteins (inflammation and lipid regulator with UBA-like and NBR1-like domains, previously C6orf106) from metazoans, choanoflagellates and protists and a diverse group of bacterial proteins, including bacterial transcriptional regulators belonging to the XRE family[24] (https://pfam.xfam.org/family/PF16158#tabview=tab7). Crystal structures have been determined for the FW domain of human NBR1 (PDB 4OLE by Joint Center for Structural Genomics) and the FW domain of human ILRUN[25], revealing an immunoglobulin-like β-sandwich fold. Despite the availability of these structures, the function of FW domains remains enigmatic. The FW domain of human NBR1 was

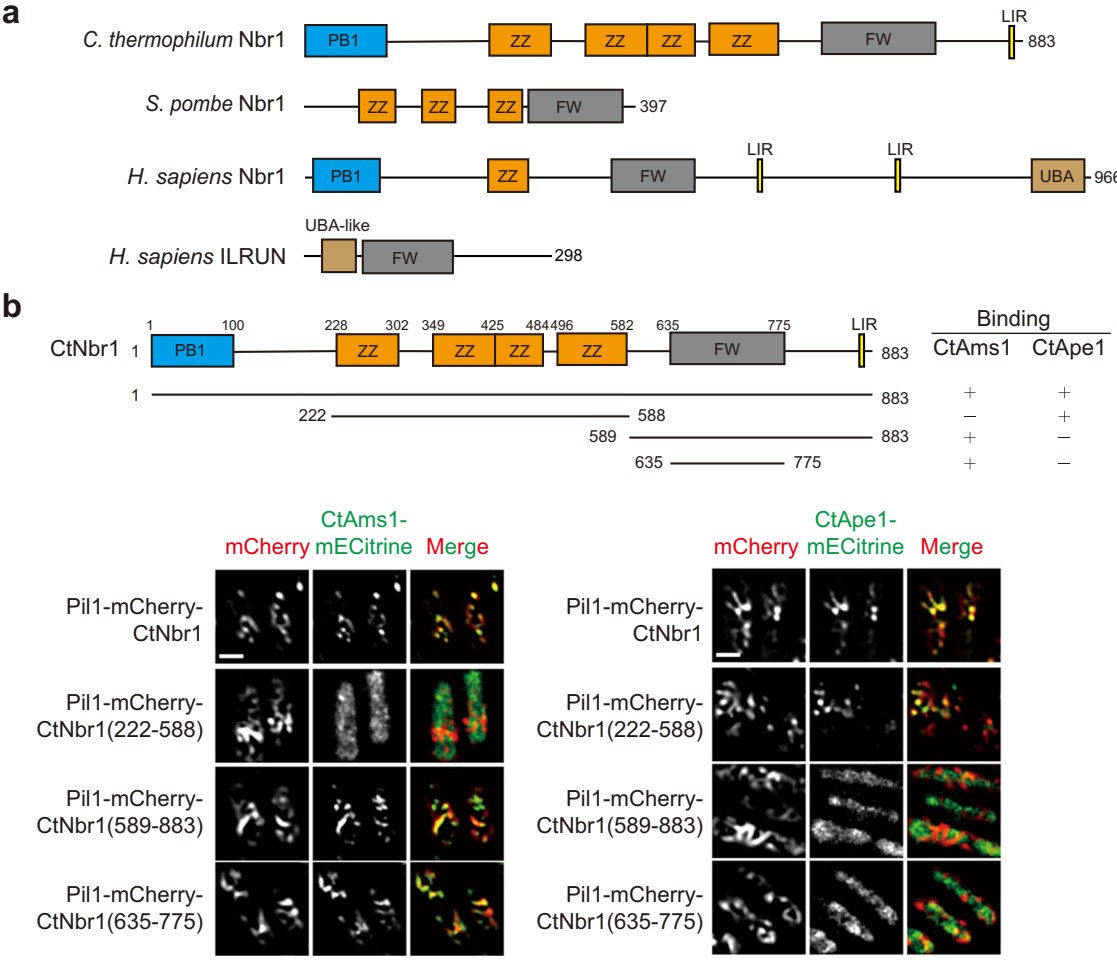

**Fig. 1 CtNbr1 binds CtAms1 and CtApe1. a** Domain architecture of NBR1 family proteins and human ILRUN. **b** Pil1 co-tethering assay. CtNbr1 and its fragments were fused to Pil1-mCherry and co-expressed with CtAms1-mECitrine or CtApe1-mECitrine in *S. pombe*. Cells were imaged by fluorescence microscopy. Bar, 3 μm. The interactions of the CtNbr1 fragments with CtAms1 and CtApe1 are summarized. The data shown are representative of two independent experiments with similar results. Source data are provided as a Source Data file.

reported to interact with the light chain of microtubule-associated protein MAP1B[24], but it is unclear whether the FW domain is important for the autophagy function of human NBR1. ILRUN was shown to interact with the transcription factor interferon regulatory factor 3 (IRF3) but it is unknown whether the FW domain is involved in this interaction[26]. Deleting the FW domain abrogated the ability of ILRUN to inhibit IRF3 signaling[25].

Ams1 is a tetrameric α-mannosidase that hydrolyzes terminal α-linked mannose in free oligosaccharides released from N-glycoproteins[27–29]. Ams1 has been shown to be transported from the cytosol into the vacuole via selective autophagy pathways in *S. cerevisiae* and *S. pombe*. In *S. pombe*, Ams1 is bound by the first ZZ domain (ZZ1) of Nbr1 and transported via the NVT pathway[15]. The ZZ1 domain in *S. pombe* Nbr1 recognizes the N-terminal peptide of Ams1 with a conserved acidic pocket and recognizes the non-N-terminal region of Ams1 with an extensive cargo-specific interface. In *S. cerevisiae*, Ams1 is bound by the autophagy receptor Atg19 or its paralog Atg34 and transported through the cytoplasm-to-vacuole targeting (Cvt) pathway that requires the conventional autophagy machinery[30]. Atg19 and Atg34 employ an Ams1-binding domain to recognize Ams1[30]. Atg19 was considered to be evolutionarily related to Nbr1 despite their low degree of amino acid identity[22].

In *S. pombe*, Nbr1 binds and transports four cargos Lap2, Ape2, Ape4, and Ams1[4,15]. To understand how conserved these Nbr1-cargo interactions are, we chose to analyze the homologs of these proteins in the filamentous fungus *Chaetomium thermophilum* (Ct), which is a thermophile increasingly used for structural biology studies[31]. We found that CtNbr1 binds CtAms1. Surprisingly, unlike *S. pombe* Nbr1, CtNbr1 binds CtAms1 using the FW domain instead of ZZ domains. We determined the structure of CtAms1 in complex with the FW domain of CtNbr1 by cryo-EM, revealing for the first time how an FW domain mediates protein-protein interaction. Combined with mutational analysis, we show that a conserved pocket of the FW domain specifically recognizes the N-terminal di-glycine of CtAms1. Our findings have implications for understanding the general binding mode of FW domains.

## Results

**CtNbr1 interacts with CtAms1 via its FW domain**. We previously showed that *S. pombe* Nbr1 (SpNbr1) uses its ZZ domains to recognize four NVT pathway cargos: Ams1, Ape4, Ape2, and Lap2[4,15]. To investigate whether this receptor-cargo relationship is conserved in *C. thermophilum*, we examined whether CtNbr1 interacts with the *Ct* homologs of NVT cargo proteins using an imaging-based Pil1 co-tethering assay[32]. In this assay, CtNbr1 was fused to the C-terminus of the Pil1-mCherry fusion protein, which localizes to distinctive filamentary structures. A cargo protein of interest was fused to the N-terminus of fluorescent protein mECitrine. The two proteins were co-expressed in *S. pombe* and if they interact with each other, co-localization on filamentary structures would be observed. We found that CtNbr1 interacted with CtAms1 (Fig. 1b), but not with CtApe4, CtApe2 or CtLap2 (Supplementary Fig. 1a). CtNbr1 possesses a PB1 domain, four ZZ domains, an FW domain, and an LIR motif (Fig. 1a). To map which domain is responsible for binding CtAms1, a series of CtNbr1 fragments were analyzed (Fig. 1b). The results showed that the FW domain of CtNbr1 mediates its interaction with CtAms1. Moreover, the interaction between the FW domain of Nbr1 and Ams1 is conserved in *Neurospora crassa* (Supplementary Fig. 1b).

We also tested whether CtNbr1 interacts with another candidate cargo protein CtApe1. The aminopeptidase Ape1 is a cargo of the Cvt pathway in *S. cerevisiae* and has a homolog in *C.*

*thermophilum* but not in *S. pombe*. We found that CtNbr1 interacted with CtApe1 by its ZZ domains (Fig. 1b).

**Structure of CtAms1-FW complex**. To reveal how the FW domain of CtNbr1 recognizes CtAms1, we determined their complex structure by cryo-EM. To stabilize the complex, a fusion protein containing CtAms1, the FW domain of CtNbr1, maltose-binding protein (MBP), and green fluorescence protein (GFP) was expressed in *S. pombe*. The GFP was used as an affinity tag and cleaved during purification (Supplementary Fig. 2a). The MBP was mobile and invisible in cryo-EM. The protein fusing strategy has been successfully applied to determine the cryo-EM structures of two complexes formed between the ZZ1 domain of SpNbr1 and the NVT cargos[15]. The micrographs were collected in a Titan Krios 300 kV EM equipped with a K2 Summit camera and an energy filter (Supplementary Fig. 2b). Following 2D and 3D classification (Supplementary Fig. 2c, d), 692,409 particles were selected to reconstruct an electron density map at an overall resolution of 2.2 Å (Fig. 2a and Supplementary Fig. 3a–e). An atomic model was built for the entire CtAms1 molecule (residues 2–1079) and the FW domain of CtNbr1 (residues 647–770) except for a disordered loop (residues 700–721) (Fig. 2b and Supplementary Fig. 3e). The density map clearly showed that the first methionine residue of CtAms1 has been processed (Supplementary Fig. 3e).

The complex structure reveals that CtAms1 forms a tetramer bound by four copies of CtNbr1 FW domain (Fig. 2a, b). The monomer structure of CtAms1 is composed of an N-terminal tail, a jelly-roll domain, a four-helix bundle, an α/β barrel, a three-helix bundle, and three β-domains (B1–B3) (Supplementary Fig. 4a). The catalytic center is located in the α/β barrel and contains a zinc ion (Supplementary Fig. 4b). The structure of CtAms1 closely resembles those of its yeast homologs (Supplementary Fig. 4c–f)[28,29]. CtAms1 and SpAms1 share 47% sequence identity and 64% sequence similarity and their monomer structures can be superimposed with a root mean square deviation (RMSD) of 0.74 Å over 898 Cα atom pairs. The oligomerization interfaces are also conserved, including the long N-terminal tail mediating extensive inter-subunit interactions[29]. Compared to SpAms1, CtAms1 contains an extension at the N-terminal tail which makes key interactions with the FW domain (Supplementary Fig. 4c).

The FW domain of CtNbr1 consists of six β-strands (β1 to β6) that form two twisted antiparallel β-sheets (Fig. 2c). β-sheet 1 is composed of β1-β3-β4 and β-sheet 2 is formed by β2-β6-β5. The two β-sheets pack into an immunoglobulin-like β-sandwich fold. The β3 and β4 strands are linked by a long and partially disordered loop that wraps the β5 and β4 strands and seals one side of the β-sandwich. The structure of the FW domain of CtNbr1 can be well aligned to those of the FW domains in human NBR1 and human ILRUN (RMSD = 0.9 and 0.93 Å, respectively)[25] (Supplementary Fig. 5a).

Each CtNbr1-FW molecule simultaneously binds two subunits of CtAms1 that are placed diagonally on one face of the tetramer structure (Figs. 2b, 3a). The interactions of the bottom right-hand molecule of CtNbr1-FW in Fig. 2b with CtAms1_3 (interface I) and CtAms1_1 (interface II) are illustrated in Fig. 3.

At interface I, one side of the CtNbr1-FW β-sandwich composed of the β3-β4 loop, the β5 strand, and the β5–β6 loop contacts the α/β barrel domain of CtAms1 (Fig. 3b). The interface is quite extensive and buries a solvent-accessible surface area of 770 Å² per molecule. The interface is of mixed nature and contains many van der Waals, hydrophobic, hydrogen bonding, and electrostatic interactions. The side-chain carboxylate of CtNbr1 D694 forms bifurcates hydrogen bonds with the

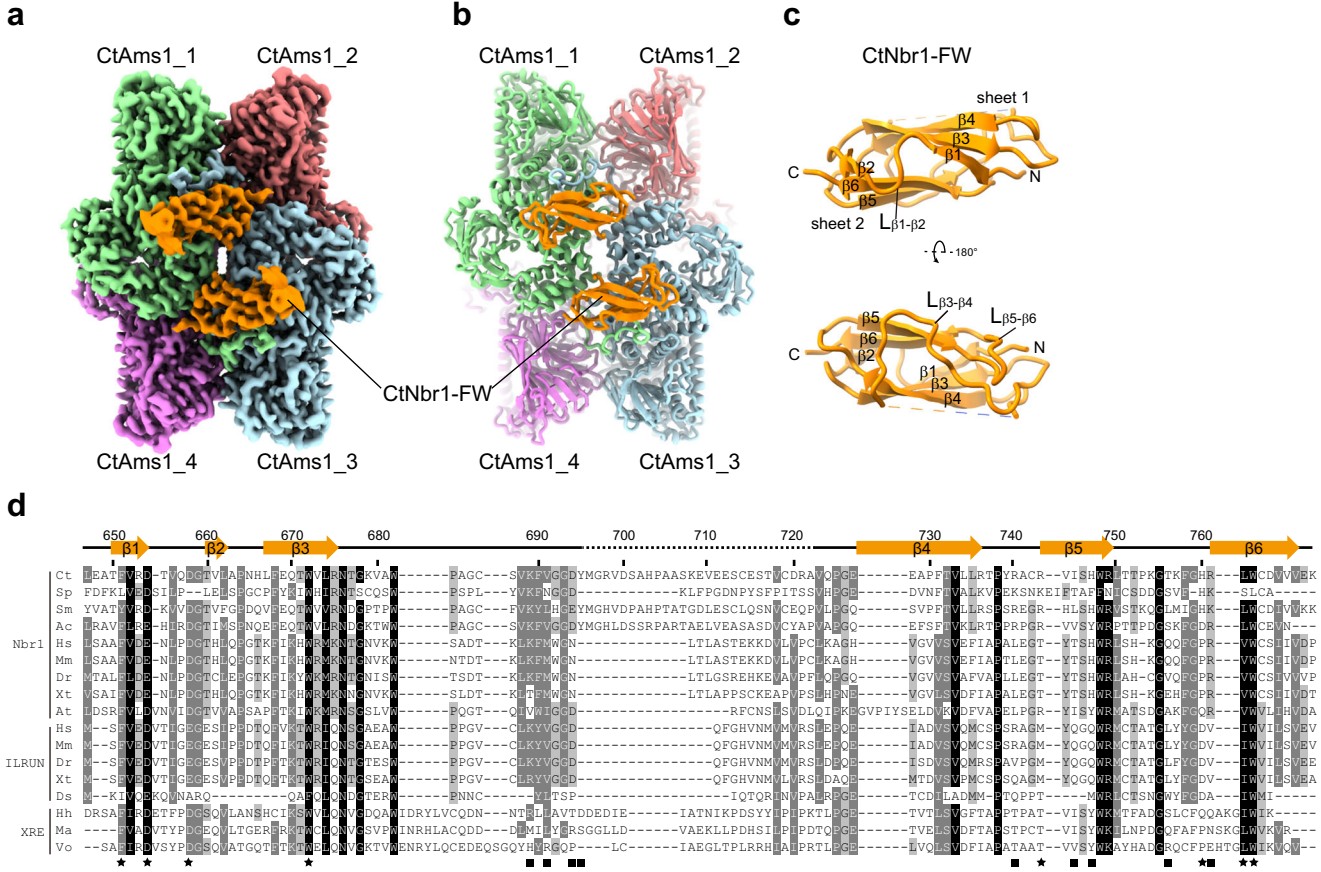

**Fig. 2 Cryo-EM structure of CtAms1 tetramer in complex with the FW domain of CtNbr1. a** Cryo-EM density map at 2.2 Å resolution. Subunits 1–4 of CtAms1 are colored in light green, pink, light cyan, and violet, and all CtNbr1 FW domains are colored in orange. **b** Ribbon representation of the CtAms1-FW complex structure. **c** Structure of the FW domain of CtNbr1. The N- and C-termini and secondary structures are labeled. **d** Multiple sequence alignment of FW domains from Nbr1, ILRUN, and bacterial XRE family proteins. These sequences are from *Chaetomium thermophilum* (Ct), *Schizosaccharomyces pombe* (Sp), *Sordaria macrospora* (Sm), *Acremonium chrysogenum* (Ac), *Homo sapiens* (Hs), *Mus musculus* (Mm), *Danio rerio* (Dr), *Xenopus tropicalis* (Xt), *Arabidopsis thaliana* (At), *Drosophila sechellia* (Ds), *Halorhodospira halochloris* (Hh), *Marinospirillum sp.* (Ma), and *Vogesella sp.* (Vo). The residues with 92%, 70%, and 50% conservation are shaded in black, gray, and light gray, respectively. The secondary structures observed in the CtNbr1-FW structure are displayed on the top. The Ams1-binding residues are marked with squares (Interface I) and stars (Interface II).

guanidine group of CtAms1 R539. Alanine mutation of either residue abolished the binding in Pil1 co-tethering assay, underscoring the importance of the interaction (Fig. 4a, b). The side chains of Y695 and R761 of CtNbr1 hydrogen bond with the backbone carbonyl groups of G280 and G356 of CtAms1, respectively. Mutational analysis showed that Y695, but not R761, is important for the binding (Fig. 4a). There are a large number of van der Waals interactions at the interface that involve P274, N275, N278, E282, P323, H525, S528, D532, and R539 of CtAms1 and K689, V691, I745, H747, and T756 of CtNbr1.

At interface II, the FW domain of CtNbr1 contacts the N-terminal tail and the 4-helix bundle of CtAms1_1 (Fig. 3c). The N-terminal tail of CtAms1_1 is anchored at the surface of CtAms1_3 before contacting β-sheet 2 of the FW domain and inserting the first two glycine residues into a pocket on the FW domain. The binding interface with the N-terminal tail of CtAms1 buries a solvent-accessible surface area of 478 Å² per molecule. In addition, Y739 of the FW domain docks on an α-helix in the 4-helix bundle of CtAms1_1 and buries a solvent-accessible surface area of 111 Å² per molecule. Mutation of Y739 to alanine did not affect the binding (Fig. 4a), suggesting that Y739 plays a minor role in binding.

The interactions between the FW domain of CtNbr1 and the CtAms1_1 tail are described below from the N14 residue of CtAms1 towards its N-terminus (Fig. 3c). The guanidine group of

CtNbr1 R743 forms hydrogen bonds with the side chain carbonyl group of CtAms1 N14 and the backbone carbonyl group of R12. The R743A mutation of CtNbr1 prevented the binding of CtAms1, whereas the N14R mutation of CtAms1 hardly affected the binding (Fig. 4a, b). These suggest that the interaction of R743 with the polypeptide backbone of CtAms1 is more crucial than the interaction with N14. The proline ring of CtAms1 P11 stacks over the tryptophan ring of CtNbr1 W763. The N-terminal tail then makes a sharp U turn at residue F6 and contacts H760 of CtNbr1 before entering the pocket.

The N-terminal dipeptide (G2-G3) of CtAms1 fits into a three-walled pocket located between the β1 and β6 strands of the FW domain of CtNbr1 (Fig. 3c, d). The bottom wall of the pocket is lined by the polypeptide backbone of the β6 strand and the side chains of L762 and W672. The two opposite side walls of the pocket are formed by the aromatic rings of residues W763 and F651. The loop between the β1 and β2 strands forms the closed end of the pocket. The pocket walls are hydrophobic, whereas the closed end of the pocket is acidic (Fig. 3e). The N-terminal dipeptide of CtAms1 pairs with the β6 strand of CtNbr1 in an antiparallel manner. Two backbone hydrogen bonds are formed between the carbonyl oxygen of G2 and the amide group of W763 (O-N distance = 2.7 Å) and between the amide group of G3 and the carbonyl oxygen of R761 (N-O distance = 3.3 Å) (Fig. 3d). All atoms of the di-glycine peptide constitute a planar structure that

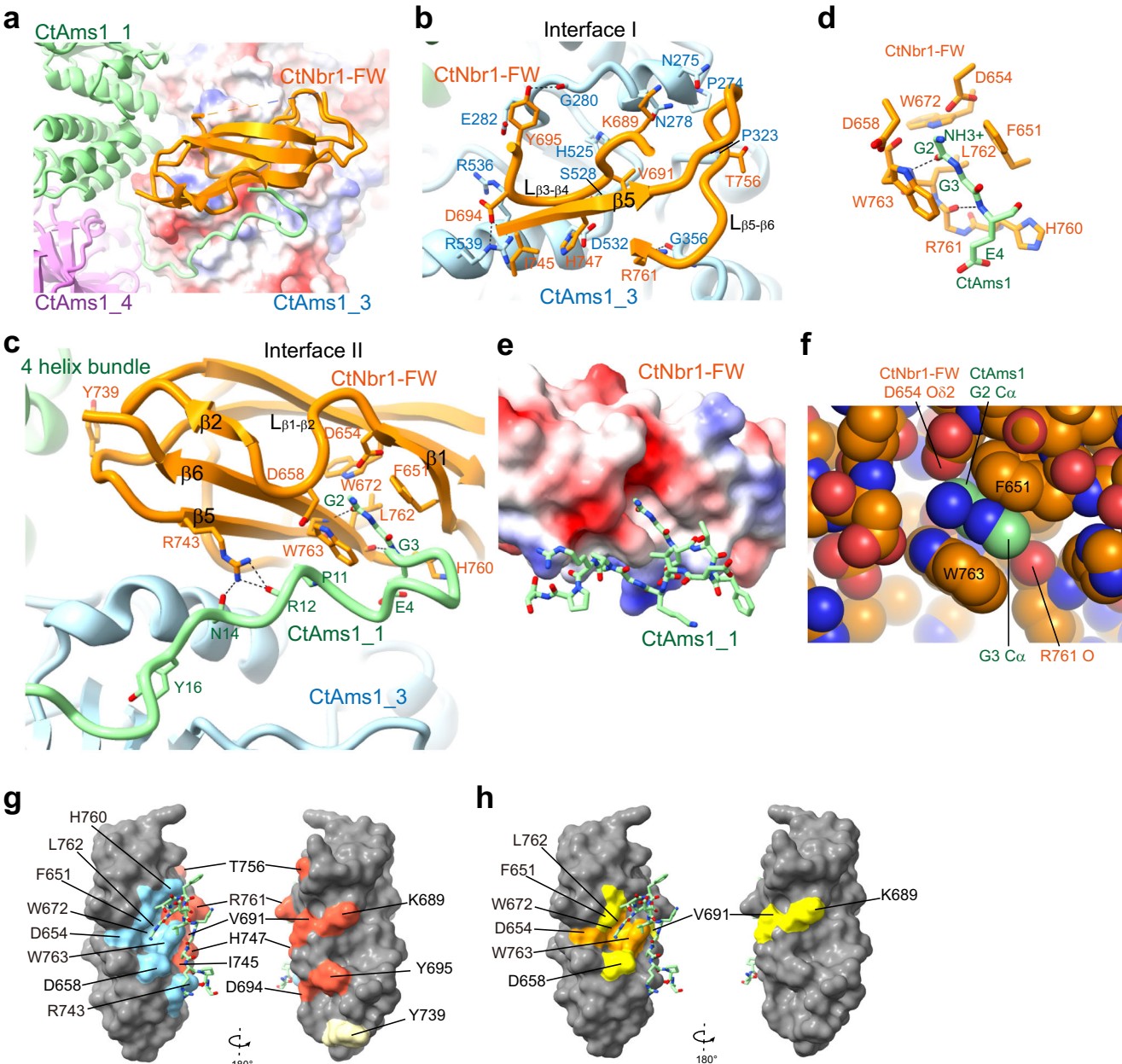

**Fig. 3 Interactions between CtAms1 and CtNbr1 FW domain. a** Overall view of the binding interfaces. The FW domain corresponds to the bottom right-hand molecule in Fig. 2a. CtAms1_3 is shown as an electrostatic potential surface. CtAms1_1 (green), CtAms1_4 (violet), and CtNbr1-FW (orange) are shown as ribbons. **b** Binding interface I between CtNbr1-FW and CtAms1_3. The interacting residues are labeled. Hydrogen bonds are denoted by black dashed lines. **c** Binding interface II between CtNbr1-FW and CtAms1_1. **d** Detailed interactions between the pocket of FW domain and the N-terminal peptide of CtAms1. **e** Electrostatic potential surface of the FW domain. The surface of the FW domain is colored in a blue-white-red gradient, with blue representing positive charge, red representing negative charge, and white representing neutral. The N-terminal peptide of CtAms1 is shown as sticks. **f** Space-filling representation of the pocket of FW domain bound to the N-terminal di-glycine of CtAms1. **g** Surface representation of the FW domain in opposite views. The residues within 4 Å of CtAms1 at the binding interface I and II are colored in brick red and cyan, respectively. Y739 was colored in light yellow. **h** Surface representation of the FW domain in opposite views. The residues with 92% and 70% conservation, as defined in Fig. 2d, are colored in orange and yellow, respectively.

is sandwiched between the aromatic rings of W763 and F651 of the FW domain. The N-terminal positively charged free amino group is within the electrostatic interaction distances of D654 and D658 of CtNbr1. Alanine mutation of F651, D654, D658, and W763 all individually abolished the interaction between CtAms1 and the FW domain of CtNbr1 in Pil1 co-tethering assay (Fig. 4a), indicating that these residues are critical for the binding.

The pocket tightly fits the two glycine residues (Fig. 3f). The Cα atom of G2 is 3.1 Å from a carboxylate oxygen atom of D654 and

the Cα atom of G3 is 3.3 Å from the carbonyl oxygen of R761. Alanine substitution of G2 or G3 both abolished the association of CtAms1 with the FW domain (Fig. 4b), indicating that the pocket cannot fit any residue with a side chain. In addition, the deletion of residues 2–5 of CtAms1 blocked the binding. These results indicate that the N-terminal peptide of CtAms1 is critical for binding the FW domain.

Residues at the pocket are highly conserved in FW domains (Figs. 2d, 3g, h and Supplementary Fig. 5b), suggesting that other

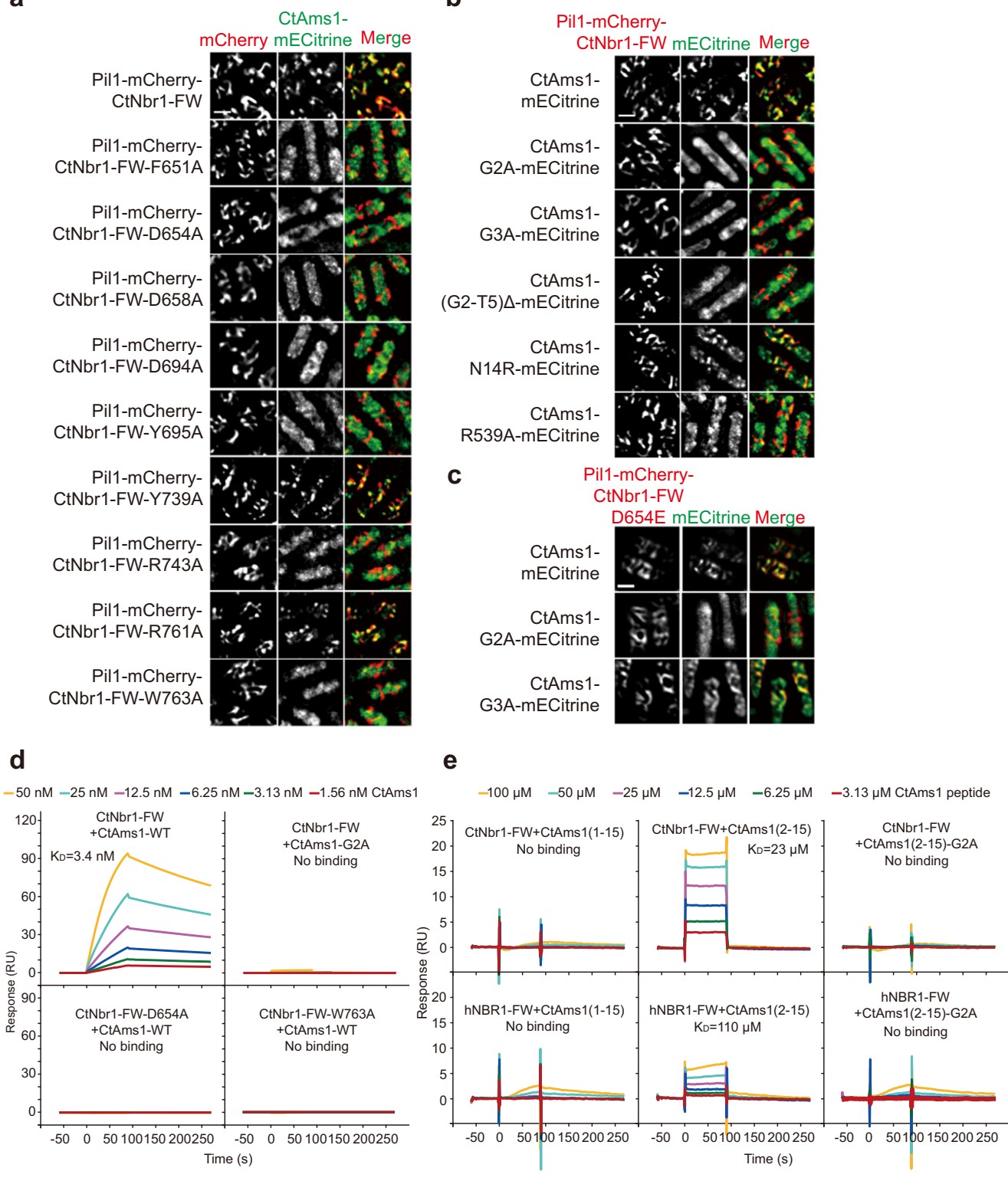

**Fig. 4 Mutational analysis of interactions between CtAms1 and CtNbr1-FW. a–c** Pil1 co-tethering assay. CtNbr1 FW domain and its mutants were fused to Pil1-mCherry and co-expressed with CtAms1-mECitrine or its mutants in *S. pombe*. Cells were imaged by fluorescence microscopy. Bar, 3 μm. The data shown are representative of two independent experiments with similar results. Source data are provided as a Source Data file. **d** SPR analysis of the CtNbr1-CtAms1 interaction. GST-tagged CtNbr1 FW domain or its mutants was bound to a CM5 sensor chip. CtAms1 or CtAms1-G2A (twofold serial dilutions from 50 nM to 1.56 nM) flowed through the chip surface. The experiments were performed once. **e** SPR analysis of the interactions of CtNbr1 and hNBR1 with CtAms1 N-terminal peptides. GST-tagged CtNbr1 FW domain or hNBR1 FW domain was bound to a CM5 sensor chip. Chemically synthesized CtAms1 N-terminal peptides (twofold serial dilutions from 100 μM to 3.13 μM) flowed through the chip surface. The experiments were performed twice with similar results.

FW domains also recognize N-terminal peptides of their ligand proteins. By contrast, other binding interfaces are not universally conserved and appear to be specific for binding Ams1.

The FW domain of CtNbr1 strictly requires glycine as the first two residues of ligand proteins, but other FW domains may have a looser requirement for the sequence of N-terminal dipeptide due to alteration of pocket residues. D654 of CtNbr1 is frequently replaced by glutamate in other FW domains particularly those in metazoan Nbr1 proteins (Fig. 2d). Glutamate should also fulfill the role of binding the N-terminal free amine group of ligand proteins, but would change the shape of the pocket. We examined the impact of glutamate substitution of D654 on the binding specificity of CtNbr1 FW domain (Fig. 4c). The pocket with D654E mutant still bound wild-type CtAms1 with an N-terminal di-glycine and strictly required glycine as the first residue of ligand (G2A mutant of CtAms1), but tolerated alanine as the second residue of ligand (G3A mutant of CtAms1). Therefore, glutamate substitution of D654 reduced the sequence specificity at the second residue of ligand.

To quantitatively measure the binding affinity between the CtNbr1-FW and CtAms1, we performed surface plasmon resonance (SPR) analysis using purified proteins and obtained a dissociation constant ($K_D$) of 3.4 nM (Fig. 4d). Consistent with the results of Pil1 co-tethering assay, mutating G2 of CtAms1 or mutating D654 or W763 of CtNbr1 abolished the interaction in the SPR assay (Fig. 4d). We also examined whether an N-terminal peptide of CtAms1 can interact with CtNbr1-FW using the SPR assay (Fig. 4e). The peptide CtAms1(2–15) bound CtNbr1-FW with a $K_D$ of 23 μM. The peptide CtAms1(1–15) failed to interact with CtNbr1-FW, indicating that the first methionine residue of CtAms1 must be removed to bind CtNbr1. As expected, mutating G2 in the peptide CtAms1(2-15) also disrupted this interaction. Because the N-terminal peptide-binding pocket is conserved in human NBR1, we analyzed whether the FW domain of human NBR1 (hNBR1-FW) can interact with the N-terminal peptide of CtAms1 (Fig. 4e). hNBR1-FW exhibited a weak binding to CtAms1(2-15) ($K_D = 110$ μM) but did not show a measurable affinity for CtAms1(1–15) or CtAms1(2-15)-G2A, indicating that hNBR1-FW also recognizes an N-terminal glycine residue.

**CtNbr1 mediates vacuolar targeting of CtAms1 in heterologous fission yeast**. Our data suggest that CtNbr1 functions as an autophagy receptor for CtAms1. Due to difficulty in performing functional studies using *C. thermophilus*, we examined the biological role of the interaction between CtNbr1 and CtAms1 in heterologous *S. pombe*. Most Nbr1 proteins have at least one LIR motif that mediates interaction with lipidated LC3/Atg8 proteins on autophagic membranes[22]. An LIR has been experimentally characterized in the Nbr1 protein of the filamentous fungus *Sordaria macrospora*[16]. CtNbr1 contains a conserved LIR motif (Fig. 1a), suggesting that it can bind Atg8 proteins. Indeed, we found that CtNbr1 can interact with *S. pombe* Atg8 (SpAtg8) in the Pil1 co-tethering assay (Fig. 5a). Under nutrient-rich conditions, CtNbr1 exhibited a diffuse cytoplasmic distribution with an accumulation at the cell periphery in *S. pombe* (Fig. 5b). The reason of this cell periphery accumulation is unclear to us. When autophagy was induced by nitrogen starvation, CtNbr1 relocalized to the vacuole lumen, and this relocalization was abolished by the absence of the core autophagy protein Atg5 (Fig. 5b), indicating that CtNbr1 can enter the vacuole through autophagy in *S. pombe*. When CtAms1 was expressed alone in *S. pombe*, it exhibited a diffuse cytoplasmic distribution under both nutrient-rich and nitrogen starvation conditions (Fig. 5c). Co-expression of CtNbr1 resulted in a cell periphery accumulation of

wild-type CtAms1 but not CtAms1-G2A or CtAms1-G3A under nutrient-rich conditions (Fig. 5c), consistent with the importance of the G2 and G3 residues of CtAms1 for the CtNbr1-CtAms1 interaction. Under nitrogen starvation conditions, co-expression of CtNbr1 resulted in the vacuolar entry of wild-type CtAms1 but not CtAms1-G2A or CtAms1-G3A (Fig. 5c). This CtNbr1-dependent vacuolar entry of CtAms1 was abolished by the absence of Atg5 (Fig. 5c). Together, these data indicate that CtNbr1 acts as an autophagy receptor for CtAms1 and undergoes conventional autophagy.

## Discussion

Nbr1 proteins are selective autophagy receptors in metazoans, fungi, and plants. The FW domain has been deemed a hallmark of Nbr1 proteins[22,23]. However, the role of the Nbr1 FW domain in selective autophagy has been unclear. In this study, we discovered that the FW domain of CtNbr1 binds CtAms1, whose homologs in budding yeast and fission yeast are selective autophagy cargos. In heterologous fission yeast, we showed that CtNbr1 can deliver CtAms1 to the vacuole via the conventional autophagy machinery. Thus, it is reasonable to propose that CtNbr1 acts as an autophagy receptor for CtAms1 and uses its FW domain for cargo recognition. Furthermore, because the key binding pocket in the FW domain of CtNbr1 is conserved in the FW domains of Nbr1 proteins in many other eukaryotes, it is possible that the ancestral function of the FW domain in Nbr1 proteins is cargo recognition.

Another type of domain ubiquitously present in Nbr1 proteins is the ZZ domain. Up to now, among Nbr1 proteins, the autophagy functions of ZZ domains have only been revealed for *S. pombe* Nbr1, whose ZZ domains are responsible for cargo binding[4,15]. In this study, we showed that CtNbr1 uses its ZZ domains to bind CtApe1, a likely cargo of selective autophagy. Thus, the cargo-binding role of ZZ domains is conserved among Nbr1 proteins.

The Cvt pathway in *S. cerevisiae* is the first discovered selective autophagy pathway and its cargos are mainly hydrolases including Ape1, Ams1, and Ape4[33–36]. These hydrolases are delivered by the Cvt pathway to the vacuole lumen to fulfill hydrolytic functions. In the fission yeast *S. pombe*, the NVT pathway performs a similar role by targeting four hydrolase cargos Ape2, Lap2, Ape4, and Ams1 to the vacuole lumen[4,15]. The selective autophagy cargo receptors of the Cvt pathway and the NVT pathway, *S. cerevisiae* Atg19 and *S. pombe* Nbr1, respectively, do not share significant sequence homology. Moreover, the membrane sequestration apparatus used by the Cvt pathway and the NVT pathway are distinct, with the former using the conventional autophagy machinery and the latter using the ESCRT machinery. Thus, it remains uncertain to what extent this type of hydrolase targeting pathway is evolutionarily conserved.

In filamentous fungi including *Aspergillus oryzae* and *Acremonium chrysogenum*, Ape1 has been shown to be transported into the vacuole in a manner dependent on the conventional autophagy machinery[37,38], suggesting the presence of a Cvt-like pathway in filamentous fungi. However, the cargo receptor for this pathway remains unknown. In our study, we found that in *C. thermophilum*, CtNbr1 binds CtApe1, suggesting that Nbr1 is the cargo receptor for Ape1 in filamentous fungi. Furthermore, the CtNbr1-CtAms1 interaction suggests that Ams1 is a conserved cargo of Nbr1 across fungi. The presence of LIR motif in CtNbr1 and its ability to target CtAms1 to the vacuole using the fission yeast's autophagy machinery suggest the presence of a Cvt-like pathway in *C. thermophilum*. In the future, it will be interesting to study whether Nbr1 proteins in non-fungal eukaryotes also mediate hydrolase targeting.

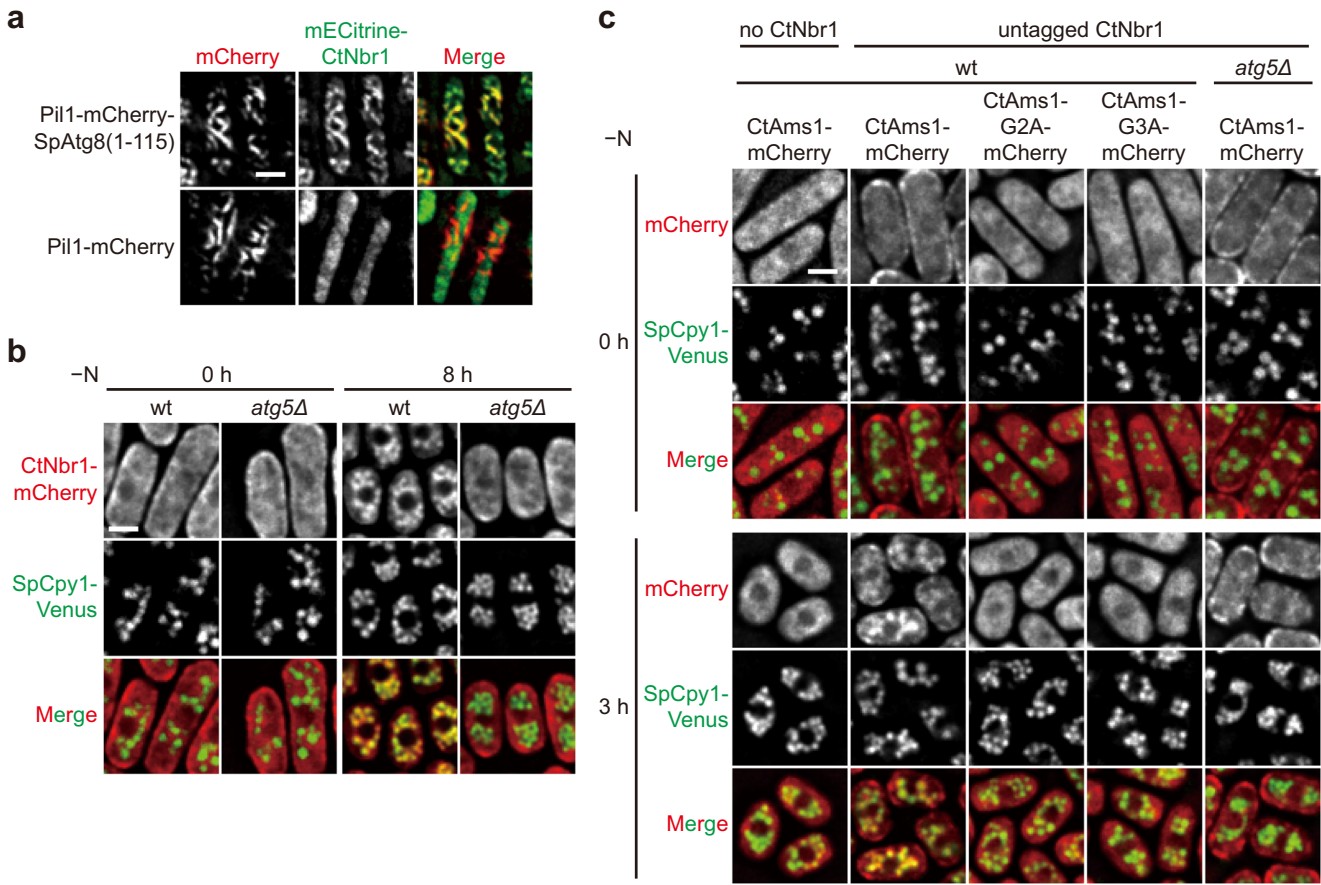

**Fig. 5 CtNbr1 mediates vacuolar targeting of CtAms1 via conventional autophagy machinery in *S. pombe*. a** CtNbr1 interacts with SpAtg8 in Pil1 co-tethering assay. SpAtg8(1−115), which lacks the G116 residue required for lipidation, was fused to Pil1-mCherry and co-expressed with mECitrine-tagged CtNbr1 in *S. pombe*. Pil1-mCherry served as a negative control. Cells were observed by fluorescence microscopy. Bar, 3 μm. **b** CtNbr1 can enter the vacuole through autophagy in nitrogen-starved *S. pombe* cells. CtNbr1-mCherry was expressed in wild-type and *atg5Δ* cells. SpCpy1-Venus was used as a vacuole lumen marker. Cells were collected before (0 h) and 8 h after nitrogen starvation (-N) and examined by fluorescence microscopy. Bar, 3 μm. **c** CtNbr1 can promote the vacuolar entry of CtAms1 in nitrogen-starved *S. pombe* cells in an autophagy-dependent manner. C-terminally mCherry-tagged CtAms1, CtAms1-G2A, and CtAms1-G3A were individually co-expressed with untagged CtNbr1 in wild-type *S. pombe* cells. CtAms1-mCherry alone expressed in wild-type cells served as a negative control. CtAms1-mCherry was also co-expressed with untagged CtNbr1 in *atg5Δ* cells. Cells were collected before (0 h) and 3 h after nitrogen starvation and were examined by fluorescence microscopy. Bar, 3 μm. The data shown are representative of two independent experiments with similar results. Source data are provided as a Source Data file.

We have determined the cryo-EM structure of the FW domain of CtNbr1 in a complex of CtAms1 tetramer. This structure reveals that CtNbr1-FW recognizes both the structure body of one CtAms1 subunit and the N-terminal tail of another CtAms1 subunit. Such a dual binding mode indicates that the FW domain of CtNbr1 recognizes the quaternary structure of CtAms1 tetramer.

Our structure reveals a partly hydrophobic and partly acidic pocket of the FW domain that fits the N-terminal di-glycine of CtAms1. The high degree of conservation of the pocket suggests that recognition of the N-terminal dipeptide of ligand proteins is the conserved protein-binding mode of the FW domain. The FW domain of human Nbr1 exhibited a weak yet detectable binding for an N-terminal peptide of CtAms1 and specificity for the N-terminal glycine. The specificity for N-terminal glycine would facilitate the search of ligand proteins for the human Nbr1 FW domain. Nevertheless, the sequence specificity of dipeptide may be different for other FW domains whose pockets are altered by conserved substitutions. We have shown that glutamate substitution of D654 would allow the pocket to accommodate alanine residue with a small side chain at the second position of the ligand.

Despite having very different structures, the FW domain is somehow similar to the ZZ domain in protein recognition mode[15,18–21]. The two domains both recognize N-terminal peptides. Their ligand binding modes both involve antiparallel strand pairing with ligand proteins and interactions with the positively charged free amine group of N-terminal peptides by conserved acidic residues. Finally, FW and ZZ domains can also bind ligand proteins with additional and extensive cargo-specific interfaces at least in the cases of Nbr1 proteins.

SpNbr1 contains an FW domain whose function remains unclear. Notably, several key residues (F651, R743, L762, W763 in CtNbr1) involved in Ams1 binding are not conserved in the FW domain of SpNbr1 (Fig. 2d). The Ams1-binding role is instead played by the ZZ1 domain of SpNbr1[15]. This transfer of the Ams1-binding role to a different domain may have caused the degeneration of the SpNbr1 FW domain. Our findings underscore the evolutionary plasticity of the cargo-binding mechanisms of autophagy receptors.

## Methods

**Fission yeast strains and plasmids**. Fission yeast strains used in this study are listed in Supplementary Table 1 and plasmids are listed in Supplementary Table 2.

Genetic methods of strain construction and composition of media were as described[39]. The coding sequences of CtNbr1 (CTHT_0062850), CtAms1 (CTHT_0067100), CtApe4 (CTHT_0070450), CtApe2 (CTHT_0067310), CtLap2 (CTHT_0062520) and CtApe1 (CTHT_0065300) were amplified from the genomic DNA of *C. thermophilum*, with introns removed by overlap PCR. We noticed that, compared to its closely related homologs, the currently annotated CtNbr1 protein sequence (NCBI accession number XP_006696593.1) is missing 29 conserved amino acids in the first ZZ domain. Inspecting the gene sequence and annotation revealed that the 87-bp coding sequence of these 29 amino acids ends with a stop codon (TAG) and is currently annotated as an intron (intron 2). By sequencing the genomic DNA, we found that there is a C-to-T sequence error in the genomic sequence and the TAG codon should be a CAG codon coding for glutamine. Using the corrected genomic sequence to perform gene structure prediction, the 87-bp sequence was no longer predicted to be an intron, but rather part of an exon.

Plasmids expressing proteins with mECitrine/mCherry/GFP tags under the control of *Pnmt1* or *P41nmt1* promoter were constructed using modified pDUAL vectors[40]. For the Pil1 co-tethering assay, the full-length and truncated Nbr1 protein was fused to the C-terminus of Pil1-mCherry and expressed under the *P41nmt1* promoter. All point mutations were generated by PCR-based mutagenesis.

The pDUAL plasmid expressing the CtAms1-CtNbr1 fusion protein under the *Pnmt1* promoter was constructed in a manner similar to the plasmid expressing the SpAms1-SpNbr1 fusion protein[15]. The DNA sequences encoding full-length CtAms1, a 30-residue linker (GGGGSGGGFKKASSSDNKEQGGGGSGGGSG), residues 635–775 of CtNbr1, a 13-residue linker (GFKKASSSDNKEQ) and MBP were assembled by overlap PCR. The PCR product was inserted into a modified pDUAL vector and placed in-frame upstream of the sequences encoding tandem cleavage sites of human rhinovirus (HRV) 3C protease and GFP. The pDUAL plasmids were linearized with NotI digestion and integrated at the *leu1* locus of the *S. pombe* genome or linearized with MluI digestion and integrated at the *ars1* locus of the *S. pombe* genome.

Plasmids expressing mECitrine- or mCherry-tagged CtNbr1 under the control of the *P41nmt1* promoter and plasmids expressing mCherry-tagged CtAms1 under the control of the *P41nmt1* promoter were constructed using modified stable integration vectors[41]. Plasmids were linearized by NotI digestion and integrated at the *ura4* or *ade6* locus of the *S. pombe* genome.

**Fluorescence microscopy.** Fission yeast cells were grown to mid-log phase in EMM medium at 30 °C. Microscopy was performed using the DeltaVision PersonalDV system (Applied Precision) equipped with a mCherry/YFP filter set (Chroma 89006 set) and a Photometrics Evolve 512 EMCCD camera. Images were obtained with a 100 × 1.4NA objective and analyzed with the SoftWoRx software.

**Pil1 co-tethering assay.** To examine a pair-wise protein-protein interaction, a bait protein was fused to Pil1-mCherry, and a prey protein was fused to mECitrine. Cells co-expressing both proteins were grown to mid-log phase in EMM medium at 30 °C for fluorescence microscopy. To image the plasma membrane-associated filament-like structures formed by a Pil1-fused protein and its interactor, we acquired 8–10 optical Z-sections so that either the top or bottom plasma membrane is in focus in one of the Z-sections. Images were processed using deconvolution using the SoftWoRx software.

**Surface plasmon resonance.** Wild-type and mutant forms of GST-CtNbr1(635–775) and wild-type GST-hNBR1(358-498) were purified from *E. coli*, and FLAG$_2$-His$_6$ (FFH)-tagged CtAms1 and CtAms1-G2A were purified from *S. pombe*. CtAms1 N-terminal peptides were chemically synthesized by Genscript. SPR analysis was performed on a Biacore T200 instrument (GE Healthcare) at 25 °C. Anti-GST antibody from the GST Capture Kit (GE Healthcare) was immobilized on a CM5 sensor chip (GE Healthcare) following the manufacturer's instruction. GST-CtNbr1(635–775) and GST-hNBR1(358–498) were diluted in running buffer (10 mM HEPES, pH 7.4. 150 mM NaCl, 0.05% Tween 20) and captured on the chip surface to about 200 response units (RU) for protein-protein interaction analysis, and about 700 RU for protein-peptide interaction analysis. Twofold serially diluted CtAms1 protein (from 50 to 1.56 nM) or peptide (from 100 to 3.13 μM) in running buffer flowed through the chip surface. The dissociation constants ($K_D$) were calculated by fitting the sensorgrams to a 1:1 binding model (for protein–protein interaction) or a steady-state model (for protein-peptide interaction) using Biacore T200 evaluation software.

**Protein preparation for cryo-EM analysis.** Fission yeast cells expressing the CtAms1-CtNbr1 FW-MBP-GFP fusion protein from the *Pnmt1* promoter were grown to mid-log phase in EMM medium at 30 °C. About 2000 OD$_{600}$ units of cells were harvested and washed once with water. Cells were lysed by grinding in liquid nitrogen. The resulting powder was mixed with lysis buffer (50 mM HEPES-Na, pH 7.5, 150 mM NaCl, 1 mM EDTA, 1 mM dithiothreitol, 1 mM phenylmethylsulfonyl fluoride, 0.05% NP-40, 10% glycerol, 1× Roche protease inhibitor cocktail). After centrifugal clarification, the cell lysate was incubated with GFP-Trap Sepharose beads (ChromoTek) for 3 h at 4 °C. The beads were washed 4 times with wash buffer (50 mM HEPES-Na, pH 7.5, 150 mM NaCl, 1 mM EDTA, 1 mM dithiothreitol, 0.05% NP-40, 10% glycerol) and incubated with 100 μl of lysis

buffer and 2 μg of 3 C protease at 4 °C overnight. The released CtAms1-CtNbr1 FW-MBP fusion protein was concentrated and buffer-exchanged to storage buffer (50 mM Tris-HCl, pH 7.5, 150 mM NaCl, 5 mM MgCl$_2$) by using an Amicon Ultra-0.5 centrifugal filter with 30 kDa molecular weight cutoff (Millipore).

**Cryo-EM data collection and processing.** To prepare vitrified specimens, three microliters of the sample (0.5 mg/ml) were placed on a glow-discharged Cryo-Matrix R1.2/1.3 300-mesh amorphous alloy film (Product no. M024-Au300-R12/13, Zhenjiang Lehua Technology Co. Ltd., China) in an FEI Vitrobot chamber, blotted for 3 sec at 100% relative humidity at 4 °C and rapidly plunged into liquid ethane. Frozen grids were stored in liquid nitrogen for data collection.

Samples were imaged on a 300 kV Titan Krios (FEI) equipped with a K2 Summit camera and a GIF energy filter (Gatan) set to a slit size of 20 eV. High-magnification images were collected in the super-resolution mode with a pixel size of 0.52 Å and then binned to a physical pixel size of 1.04 Å. A total of 2925 micrographs were collected using SerialEM[42]. Each micrograph composed of 32 frames was collected with a total electron dose of ~50 eÅ$^{-2}$.

The beam-induced motion was corrected using MotionCor2[43]. Parameters of contrast transfer function (CTF) were estimated with Gctf[44]. Particles were automatically picked by gautomatch (http://www.mrclmb.cam.ac.uk/kzhang/Gautomatch/). The particles were initially extracted with 4 × 4 binning and subjected to two rounds of 2D classification in CryoSPARC v2.05[45]. A total of 822,430 particles were selected for 3D classification using Relion 3.1[46]. The cryo-EM density map of *S. pombe* Ams1 (EMD-30021) was used as the initial model[29]. In all, 760,890 particles were selected and re-imported into CryoSPARC for heterogeneous refinement with D2 symmetry. Finally, 692,409 particles were selected and re-extracted with the original pixel size of 1.04 Å and a box size of 320 pixels. These particles were subjected to non-uniform refinement in CryoSPARC with D2 symmetry and the option of CTF refinement (Supplementary Fig. 2d). A map was reconstructed at 2.2 Å resolution based on the Fourier shell correlation (FSC) = 0.143 criterion (Supplementary Fig. 3c). The map was further post-processed with LocScale[47], yielding a better-resolved density for CtNbr1 (Supplementary Fig. 3a).

**Model building and refinement.** The initial homology models for CtAms1 and CtNbr1 were generated by SWISS-MODEL using the structures of SpAms1 (PDB code: 6LZ1) and the FW domain of human NBR1 (PDB code: 4OLE) as templates[48]. The structural model was manually adjusted according to the LocScale map in COOT[49]. The model was refined in real space using PHENIX against the LocScale map[50]. The refinement parameters included global minimization, local grid search, atomic displacement parameters refinement, secondary structure restraints, a nonbonded weight of 4, and non-crystallographic symmetry restrains. The current model contains residues 2–1079 of CtAms1 and residues 647–699 and 722–770 of CtNbr1. The map-to-model FSC curve was calculated in PHENIX.

Structural figures were prepared with ChimeraX[51] and Chimera[52]. The buried accessible surface area was calculated with areaimol using a probe of 1.4 Å radius[53]. Cryo-EM data collection and refinement statistics are summarized in Supplementary Table 3.

**Reporting summary.** Further information on research design is available in the Nature Research Reporting Summary linked to this article.

## Data availability
The cryo-EM density map and coordinates have been deposited to the Electron Microscopy Data Bank (EMDB) and Protein Data Bank (PDB) under accession numbers EMD-32091 and 7VQO. The correct coding sequence of CtNbr1 has been deposited to GenBank under the accession number MZ729680. The authors declare that all other data supporting the findings of this study are available within the paper and its supplementary information files. Source data are provided in this paper.

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

## Acknowledgements
We thank Boling Zhu, Lihong Chen, Xiaojun Huang, Fei Sun, and other staff at the Center for Biological Imaging, Institute of Biophysics, Chinese Academy of Sciences for their assistance with cryo-EM data collection. We thank Dan-Dan Xu for providing the genomic DNA of *C. thermophilum*, Fang Suo for help in bioinformatics analysis, and other members of the laboratories for discussion. This work was supported by the National Natural Science Foundation of China [32071199, 91940302], the Strategic Priority Research Program of the Chinese Academy of Sciences [XDB37010201], and the National Key R&D Program of China [2017YFA0504600] to K.Y. and grants from the Chinese Ministry of Science and Technology and the Beijing municipal government to L.-L.D.

## Author contributions
J.Z., Y.-Y.W., Z.-Q.P, L.-L.D., and K.Y. designed the work. J.Z. determined the cryo-EM structure. Y.-Y.W. and Z.-Q.P conducted yeast studies and prepared proteins. Y.L. performed SPR analyses. J.S., L.-L.D, and K.Y. supervised the work. All authors contributed to the manuscript preparation.

## Competing interests
The authors declare no competing interests.
