## [Peer Review File · Nature Communications]

Structural mechanism of protein recognition by the FW domain of autophagy receptor Nbr1REVIEWER COMMENTS

Reviewer #1 (Remarks to the Author):

Selective autophagy contributes to cellular homeostasis through selective degradation of various cytoplasmic materials including proteins and organelles. During this process, cargos are specifically recognized by cargo receptors, which also interact with lipidated Atg8 on autophagic membranes using a LIR/AIM motif, thereby enabling selective sequestration of the cargos into autophagosomes. Nbr1 is one of such autophagy receptors conserved across eukaryotes. Mammalian and plant NBR1 target protein aggregates for autophagic degradation, whereas in fission yeast, it transports hydrolases including Ams1 to the vacuole through ESCRT-mediated vacuolar targeting (NVT) pathway. The FW domain is a small globular domain possessing an immunoglobulin-like beta-sandwich fold and is found in all Nbr1 proteins; however, its role in selective autophagy remained elusive. In order to understand how Nbr1 recognizes its cargos, the authors studied the interaction of filamentous fungus Nbr1 (CtNbr1) with CtAms1 and found that the FW domain of CtNbr1 binds CtAms1, which is in contrast to fission yeast Nbr1 that binds Ams1 using the ZZ domain. Moreover, the authors determined the cryo-EM structure of the CtAms1-CtNbr1 FW complex and, together with mutational analyses, unveiled the detailed interaction mode of the CtNbr1 FW domain with CtAms1.

The structural study is solid and clearly reveals the detailed interaction between CtAms1 and the FW domain of CtNbr1. On the other hand, functional studies are limited to the Pil1 co-tethering assay, which simply studies the CtAms1-CtNbr1 interaction qualitatively. This manuscript contains two big shortcomings: one is lack of any data showing biological significance of the CtAms1-CtNbr1 FW interaction, and another is lack of information about the generality of the established CtAms1-CtNbr1 FW interaction. Without these data, this manuscript is more appropriate for specific journals because it simply describes the detailed interaction mode between CtAms1 and CtNbr1 FW.

Major points

1) It has not been demonstrated that CtAms1 is a selective cargo for autophagy and that CtNbr1 functions as a receptor for the process. In the case of fission yeast, Ams1 is delivered to the vacuole via NVT pathway, in which Nbr1 functions as a receptor and binds Ams1 using the ZZ domain. Because CtNbr1 uses the FW domain rather than the ZZ domain, it might be possible that CtNbr1 FW-CtAms1 interaction has some role(s) other than selective autophagy or NVT pathway. Study whether CtNbr1 FW can function as a receptor for CtAms1. This reviewer understands that it is difficult for the authors to perform functional studies using *C. thermophilum*. Instead, express CtAms1 and CtNbr1 in fission yeast and analyze whether CtAms1 can be a selective cargo for the NVT pathway. CtNbr1 may not function as a receptor in fission yeast and so it would be better to use a fission yeast Nbr1 mutant whose FW domain is replaced with that of CtNbr1. Alternatively, budding yeast can be used, and in that case express Atg19 mutant that possesses the FW domain of CtNbr1 and analyze whether CtAms1 expressed in budding yeast is delivered to the vacuole via selective autophagy (that includes Cvt pathway).

2) The interaction between CtAms1 and CtNbr1 was studied solely by Pil1 co-tethering assay. Pil1 co-tethering assay seems to have worked well in this manuscript, but it provides only qualitative information. It is unclear how strong CtAms1 binds to CtNbr1 and what magnitude each mutation affects the interaction. Perform a quantitative binding assay (such as ITC or SPR) for wild-type and some important mutants of CtAms1-CtNbr1. Moreover, the affinity of the N-terminal peptide including di-glycine from CtNbr1 to CtAms1 should also be analyzed.

3) This is the first manuscript that describes the structural basis of the FW domain-mediated protein recognition. The authors revealed the specific recognition of the N-terminal di-glycine by the FW domain of CtNbr1. It is interesting whether this specific recognition by the FW domain is evolutionarily conserved or specific to CtNbr1. Study whether other FW domains (such as hNBR1 FW or hLRUN FW) could also bind a peptide possessing N-terminal di-glycine sequence experimentally (for example, SPR or ITC). Such information is important for increasing the generality of this work.

Minor points

1) Based on the map in Figure S3E, the authors concluded that the first methionine of CtAms1 has been processed. However, the map is difficult to see. Provide a stereo view of the map focused on the N-terminal portion of CtAms1 with a labeling of Gly2.

2) At lines 164-165, "The guanidine group of CtNbr1 R743 forms hydrogen bonds with the side chain amide group of CtAms1 N14", the guanidine group should form a hydrogen bond with a carbonyl group rather than an amide group of N14.

Reviewer #2 (Remarks to the Author):

The selective autophagy receptor NBR1 is conserved across eukaryotes and is characterized by its FW (four tryptophan) domain. Crystal structures of FW domains from the human NBR1 protein and the human ILRUN protein are available. However, how the FW domain contributes to selective cargo binding is so far unknown. Zhang et al. determined the cryo-EM structure of the FW domain from the thermophilic fungus *Chaetomium thermophilum* in complex with its cargo alpha mannosidase AMS1. The structure revealed that CtASM1 forms a tetramer and that each FW domain simultaneously binds two subunits of the tetramer structure. The FW domain binds to the structure of one ASM1 subunit and an N-terminal di-glycine motif of another subunit thereby recognizing the global structure of the ASM1 tetramer.

The work is of great importance for the field of selective autophagy and for the first time describes the binding mode of the receptor NBR1 to its cargo. The experimental work presented was carefully conducted. Only a few points should be improved.

In detail, I have the following remarks:

L. 24 Abstract: It should be mentioned that Asm1 is an alpha mannosidase

Figure 1, 4 and (Fig. S1: The Pil 1 co-tethering assay in *S. pombe* was used to analyze interaction of CtNBR1 and its putative cargo as well as the interaction mutated versions of the NBR1 FW domain and its cargo ASM1. A more rigorous quantification of the microscopy experiments would strengthen the significance and would make the experiments more comparable.

L. 39 perhaps already here it should be mentioned that *S. cerevisiae* has no clear NBR1 homolog.

L. 111: *Neurospora* instead of *Neurosporus*

L.117-118: from the text and Figure S2A it seems that CtAMS1 FW fusion was expressed and purified as a MBP protein, however, from the Methods section it seem that it is a GFP fusion protein (see L309). This should be clarified in the text and Figure S2A.

L. 130: The CtASM1 structure is only compared to ASM1 from *S. pombe* (Fig. S4) is this also true for ASM1 from *S. cerevisiae*?

Reviewer #3 (Remarks to the Author):

The autophagy receptor Nbr1 in the filamentous fungus *Chaetomium thermophilum* is here shown to use its FW domain to interact with the α -mannosidase Ams1. The same authors previously reported that Nbr1 in the fission yeast *S. pombe* uses its ZZ domain to interact with Ams1, and this interaction is needed for the vacuolar delivery of SpNbr1 via an ESCRT-dependent microautophagy pathway. The use

of the FW domain in CtNbr1 suggests an interesting evolutionary plasticity in the use of domains for cargo interactions, and the experimental data and conclusions are solid.

The described interaction suggests a role for CtNbr1 in the vacuolar targeting of CtAms1, but the authors do not here experimentally verify that CtNbr1 is responsible for the delivery of Ams1 to the vacuole. They neither test if the vacuolar targeting of Ams1 in *C. thermophilum* is by conventional macroautophagy or by an alternative ESCRT pathway as in *S. pombe*.

The interaction between the CtNbr1 FW domain and a CtAms1 tetramer is characterized using Cryo-EM. Two binding interfaces are identified, and the structure is determined at 2.2 Å resolution. The figures illustrating the interaction interfaces are very informative, and their structural data are strongly supported experimentally by the introduction of point mutations. They identify a highly conserved binding pocket in the FW domain that they suggest may represent an ancestral function of the FW domain in cargo recognition. The FW domain is a signature of non-metazoan and metazoan Nbr1 orthologues. The evidence for a use of the FW domain in cargo recognition is very interesting and has potential significance also for studies of p62 or Nbr1 mediated selective autophagy processes in mammals.

Major comment:

In the discussion, the delivery route for CtAms1 is discussed, and the authors speculate that it can be conventional macroautophagy (Cvt pathway) since this is the delivery route for Ape1 in two other filamentous fungi species. But they don't experimentally test if CtAms1 in *C. thermophilum* is degraded via the macroautophagy pathway. The mentioned studies did not include Nbr1, and we are left in confusion since SpNbr1 delivers SpAms1 via an ESCRT pathway in the fission yeast *S. pombe*. The submitted manuscript may not depend on this type of data, but fungal autophagy receptors like CtNbr1 and SpNbr1 are orthologues of metazoan p62/Nbr1, and it is of interest to characterize the autophagy pathways used by these fungal autophagy receptors. Data on this would improve the manuscript. Is it possible to use *S. pombe* as a system to test if CtNbr1 can drive the lysosomal delivery of CtAms1 (or CtApe1), and if this depends on macroautophagy?

Minor comments:

In the figure legend for figure 2 (last sentence), interface I residues are said to be marked with stars, and interphase II residues with squares. This is wrong. The opposite is correct!

In figure S5, the residue marked D659 (red) should be D658?

In the introduction (line 56), the term LC3/Atg8 to summarize all ATG8 orthologues is a bit incorrect, and maybe Atg8 orthologues or LC3/GABARAP/Atg8 is better.

Reviewer #1 (Remarks to the Author):

Selective autophagy contributes to cellular homeostasis through selective degradation of various cytoplasmic materials including proteins and organelles. During this process, cargos are specifically recognized by cargo receptors, which also interact with lipidated Atg8 on autophagic membranes using a LIR/AIM motif, thereby enabling selective sequestration of the cargos into autophagosomes. Nbr1 is one of such autophagy receptors conserved across eukaryotes. Mammalian and plant NBR1 target protein aggregates for autophagic degradation, whereas in fission yeast, it transports hydrolases including Ams1 to the vacuole through ESCRT-mediated vacuolar targeting (NVT) pathway. The FW domain is a small globular domain possessing an immunoglobulin-like beta-sandwich fold and is found in all Nbr1 proteins; however, its role in selective autophagy remained elusive. In order to understand how Nbr1 recognizes its cargos, the authors studied the interaction of filamentous fungus Nbr1 (CtNbr1) with CtAms1 and found that the FW domain of CtNbr1 binds CtAms1, which is in contrast to fission yeast Nbr1 that binds Ams1 using the ZZ domain. Moreover, the authors determined the cryo-EM structure of the CtAms1-CtNbr1 FW complex and, together with mutational analyses, unveiled the detailed interaction mode of the CtNbr1 FW domain with CtAms1.

The structural study is solid and clearly reveals the detailed interaction between CtAms1 and the FW domain of CtNbr1. On the other hand, functional studies are limited to the Pil1 co-tethering assay, which simply studies the CtAms1-CtNbr1 interaction qualitatively. This manuscript contains two big shortcomings: one is lack of any data showing biological significance of the CtAms1-CtNbr1 FW interaction, and another is lack of information about the generality of the established CtAms1-CtNbr1 FW interaction. Without these data, this manuscript is more appropriate for specific journals because it simply describes the detailed interaction mode between CtAms1 and CtNbr1 FW.

Major points

1) It has not been demonstrated that CtAms1 is a selective cargo for autophagy and that CtNbr1 functions as a receptor for the process. In the case of fission yeast, Ams1 is delivered to the vacuole via NVT pathway, in which Nbr1 functions as a receptor and binds Ams1 using the ZZ domain. Because CtNbr1 uses the FW domain rather than the ZZ domain, it might be possible that CtNbr1 FW-CtAms1 interaction has some role(s) other than selective autophagy or NVT pathway. Study whether CtNbr1 FW can function as a receptor for CtAms1. This reviewer understands that it is difficult for the authors to perform functional studies using *C. thermophilum*. Instead, express CtAms1 and CtNbr1 in fission yeast and analyze whether CtAms1 can be a selective cargo for the NVT pathway. CtNbr1 may not function as a receptor in fission yeast and so it would be better to use a fission yeast Nbr1 mutant whose FW domain is replaced with that of CtNbr1. Alternatively, budding yeast can be used, and in that case express Atg19 mutant that possesses the FW domain of CtNbr1 and analyze whether CtAms1 expressed in budding yeast is delivered to the vacuole via selective autophagy (that includes Cvt pathway).

Response: It was a great idea to test whether CtNbr1 is an autophagy receptor for CtAms1 in a heterologous yeast. We followed the reviewer's suggestions and used *S. pombe* to examine the functional significance of the CtNbr1-CtAms1 interaction. CtNbr1 contains a conserved LIR motif that is likely recognized by Atg8 in other yeasts. Indeed, we found that CtNbr1 can bind to fission yeast Atg8 and is transported into the vacuole through autophagy during nitrogen starvation. Furthermore, we found that, during nitrogen starvation, CtNbr1 can promote the relocalization of CtAms1 from the cytoplasm to the vacuole lumen in a manner dependent on the CtNbr1-CtAms1 interaction and the autophagy machinery. Thus, the interaction between CtNbr1 and CtAms1 allows CtNbr1 to function as an autophagy receptor for CtAms1 in *S. pombe*. These results are shown in Fig. 5 of the revised

manuscript.

2) The interaction between CtAms1 and CtNbr1 was studied solely by Pil1 co-tethering assay. Pil1 co-tethering assay seems to have worked well in this manuscript, but it provides only qualitative information. It is unclear how strong CtAms1 binds to CtNbr1 and what magnitude each mutation affects the interaction. Perform a quantitative binding assay (such as ITC or SPR) for wild-type and some important mutants of CtAms1-CtNbr1. Moreover, the affinity of the N-terminal peptide including di-glycine from CtNbr1 to CtAms1 should also be analyzed.

Response: We followed the suggestions of the reviewer and performed SPR to quantitatively measure the binding affinity. Consistent with the Pil1 co-tethering assay results, wild-type CtNbr1-FW can interact with wild-type CtAms1 in the SPR assay ($K_D=3.4$ nM), while mutating G2 of CtAms1 disrupted this interaction. Mutating D654 or W763 of CtNbr1-FW also disrupted the interaction. We also used the SPR analysis to examine the binding of CtAms1 N-terminal peptides to CtNbr1-FW. The peptide CtAms1(2-15) can interact with CtNbr1-FW ($K_D=23$ μ M), whereas neither CtAms1(1-15) nor CtAms1(2-15)-G2A mutant can interact with CtNbr1-FW. These results are shown in Figs. 4D-E of the revised manuscript.

3) This is the first manuscript that describes the structural basis of the FW domain-mediated protein recognition. The authors revealed the specific recognition of the N-terminal di-glycine by the FW domain of CtNbr1. It is interesting whether this specific recognition by the FW domain is evolutionarily conserved or specific to CtNbr1. Study whether other FW domains (such as hNBR1 FW or hLRUN FW) could also bind a peptide possessing N-terminal di-glycine sequence experimentally (for example, SPR or ITC). Such information is important for increasing the generality of this work.

Response: We have used SPR to examine whether the FW domain of human NBR1 (hNBR1-FW) can bind the N-terminal peptide of CtAms1. hNBR1-FW exhibited a weak yet detectable binding for the peptide CtAms1(2-15) ($K_D=110$ μ M) but showed no measurable affinity for CtAms1(1-15) and CtAms1(2-15)-G2A, indicating that the ligand binding mode of CtNbr1-FW is likely conserved in hNBR1-FW. These results are shown in Figs. 4E of the revised manuscript.

Minor points

1) Based on the map in Figure S3E, the authors concluded that the first methionine of CtAms1 has been processed. However, the map is difficult to see. Provide a stereo view of the map focused on the N-terminal portion of CtAms1 with a labeling of Gly2.

Response: We have added a new panel in Fig. S3E to highlight the density of CtAms1 residues G2-T5. As the density map is rather simple, stereo views are not used.

2) At lines 164-165, "The guanidine group of CtNbr1 R743 forms hydrogen bonds with the side chain amide group of CtAms1 N14", the guanidine group should form a hydrogen bond with a carbonyl group rather than an amide group of N14.

Response: corrected to "the side chain carbonyl group of CtAms1 N14".

Reviewer #2 (Remarks to the Author):

The selective autophagy receptor NBR1 is conserved across eukaryotes and is characterized by its FW (four

tryptophan) domain. Crystal structures of FW domains from the human NBR1 protein and the human ILRUN protein are available. However, how the FW domain contributes to selective cargo binding is so far unknown. Zhang et al. determined the cryo-EM structure of the FW domain from the thermophilic fungus *Chaetomium thermophilum* in complex with its cargo alpha mannosidase AMS1. The structure revealed that CtASM1 forms a tetramer and that each FW domain simultaneously binds two subunits of the tetramer structure. The FW domain binds to the structure of one ASM1 subunit and an N-terminal di-glycine motif of another subunit thereby recognizing the global structure of the ASM1 tetramer.

The work is of great importance for the field of selective autophagy and for the first time describes the binding mode of the receptor NBR1 to its cargo. The experimental work presented was carefully conducted. Only a few points should be improved.

In detail, I have the following remarks:

L. 24 Abstract: It should be mentioned that Asm1 is an alpha mannosidase

Response: This point has been already mentioned in the Abstract: "Here, we show that Nbr1 from the filamentous fungus *Chaetomium thermophilum* uses its FW domain to bind the α -mannosidase Ams1, a cargo of selective autophagy in both budding yeast and fission yeast."

Figure 1, 4 and Fig. S1: The Pil 1 co-tethering assay in *S. pombe* was used to analyze interaction of CtNBR1 and its putative cargo as well as the interaction mutated versions of the NBR1 FW domain and its cargo ASM1. A more rigorous quantification of the microscopy experiments would strengthen the significance and would make the experiments more comparable.

Response: We have performed SPR analysis to quantitatively measure the binding affinity. The results are shown in Figs. 4D-E of the revised manuscript.

L. 39 perhaps already here it should be mentioned that *S. cerevisiae* has no clear NBR1 homolog.

Response: The sentence has been changed to "Nbr1 is an autophagy receptor conserved across eukaryotes, but notably absent in *S. cerevisiae*."

L. 111: *Neurospora* instead of *Neurosporus*

Response: Corrected.

L.117-118: from the text and Figure S2A it seems that CtAMS1 FW fusion was expressed and purified as a MBP protein, however, from the Methods section it seem that it is a GFP fusion protein (see L309). This should be clarified in the text and Figure S2A.

Response: The initial fusion protein did contain a GFP tag at the C-terminus. The GFP tag was bound to GFP-Trap beads and cleaved during purification. The resulting CtAms1-FW-MBP fusion was shown in the SDS-PAGE (Fig. S2A) and used for cryo-EM analysis. The description of the fusion protein has been revised in the text.

L. 130: The CtASM1 structure is only compared to ASM1 from *S. pombe* (Fig. S4) is this also true for ASM1 from *S. cerevisiae*?

Response: Structural alignment between CtAms1 and ScAms1 has been added in Fig. S4E-F. The overall structures of SpAms1, ScAms1 and CtAms1 are all similar. The cryo-EM structure of ScAms1 was determined at 6.3 Å resolution and its entire long N-terminal tail was not modeled.

Reviewer #3 (Remarks to the Author):

The autophagy receptor Nbr1 in the filamentous fungus *Chaetomium thermophilum* is here shown to use its FW domain to interact with the α -mannosidase Ams1. The same authors previously reported that Nbr1 in the fission yeast *S. pombe* uses its ZZ domain to interact with Ams1, and this interaction is needed for the vacuolar delivery of SpNbr1 via an ESCRT-dependent microautophagy pathway. The use of the FW domain in CtNbr1 suggests an interesting evolutionary plasticity in the use of domains for cargo interactions, and the experimental data and conclusions are solid.

The described interaction suggests a role for CtNbr1 in the vacuolar targeting of CtAms1, but the authors do not here experimentally verify that CtNbr1 is responsible for the delivery of Ams1 to the vacuole. They neither test if the vacuolar targeting of Ams1 in *C. thermophilum* is by conventional macroautophagy or by an alternative ESCRT pathway as in *S. pombe*.

The interaction between the CtNbr1 FW domain and a CtAms1 tetramer is characterized using Cryo-EM. Two binding interfaces are identified, and the structure is determined at 2.2 Å resolution. The figures illustrating the interaction interfaces are very informative, and their structural data are strongly supported experimentally by the introduction of point mutations. They identify a highly conserved binding pocket in the FW domain that they suggest may represent an ancestral function of the FW domain in cargo recognition. The FW domain is a signature of non-metazoan and metazoan Nbr1 orthologues. The evidence for a use of the FW domain in cargo recognition is very interesting and has potential significance also for studies of p62 or Nbr1 mediated selective autophagy processes in mammals.

Major comment:

In the discussion, the delivery route for CtAms1 is discussed, and the authors speculate that it can be conventional macroautophagy (Cvt pathway) since this is the delivery route for Ape1 in two other filamentous fungi species. But they don't experimentally test if CtAms1 in *C. thermophilum* is degraded via the macroautophagy pathway. The mentioned studies did not include Nbr1, and we are left in confusion since SpNbr1 delivers SpAms1 via an ESCRT pathway in the fission yeast *S. pombe*. The submitted manuscript may not depend on this type of data, but fungal autophagy receptors like CtNbr1 and SpNbr1 are orthologues of metazoan p62/Nbr1, and it is of interest to characterize the autophagy pathways used by these fungal autophagy receptors. Data on this would improve the manuscript. Is it possible to use *S. pombe* as a system to test if CtNbr1 can drive the lysosomal delivery of CtAms1 (or CtApe1), and if this depends on macroautophagy?

Response: We have shown in fission yeast that CtNbr1 can deliver CtAms1 to the vacuole through the macroautophagy pathway (Fig. 5). Please see our response to question 1 of reviewer #1.

Minor comments:

In the figure legend for figure 2 (last sentence), interface I residues are said to be marked with stars, and interphase II residues with squares. This is wrong. The opposite is correct!

Response: Corrected.

In figure S5, the residue marked D659 (red) should be D658?

Response: Corrected.

In the introduction (line 56), the term LC3/Atg8 to summarize all ATG8 orthologues is a bit incorrect, and maybe Atg8 orthologues or LC3/GABARAP/Atg8 is better.

Response: Corrected.

REVIEWERS' COMMENTS

Reviewer #1 (Remarks to the Author):

The authors have addressed all of my concerns and the manuscript has improved dramatically.

Reviewer #2 (Remarks to the Author):

All reviewer suggestions for improvement were integrated in the revised version of the manuscript, which is now considerably improved.

I have no further comments.

Reviewer #3 (Remarks to the Author):

In the revised manuscript, the authors have successfully responded to all the issues I had. In particular, the new data that CtAms1 is degraded by autophagy if co-expressed with CtNbr1 in *S. pombe* strongly improves the significance of the manuscript.